Ecological and Evolutionary Science

# Space Is More Important than Season when Shaping Soil Microbial Communities at a Large Spatial Scale

Kaoping Zhang,[a,b] Manuel Delgado-Baquerizo,[c] Yong-Guan Zhu,[d,e] Haiyan Chu[a,b]

[a]State Key Laboratory of Soil and Sustainable Agriculture, Institute of Soil Science, Chinese Academy of Sciences, Nanjing, China
[b]University of Chinese Academy of Sciences, Beijing, China
[c]Departamento de Sistemas Físicos, Químicos y Naturales, Universidad Pablo de Olavide, Seville, Spain
[d]Key Laboratory of Urban Environment and Health, Institute of Urban Environment, Chinese Academy of Sciences, Xiamen, China
[e]State Key Lab of Urban and Regional Ecology, Research Centre for Eco-environmental Sciences, Chinese Academy of Sciences, Beijing, China

**ABSTRACT** The relative importance of spatial and temporal variability in shaping the distribution of soil microbial communities at a large spatial scale remains poorly understood. Here, we explored the relative importance of space versus time when predicting the distribution of soil bacterial and fungal communities across North China Plain in two contrasting seasons (summer versus winter). Although we found that microbial alpha (number of phylotypes) and beta (changes in community composition) diversities differed significantly between summer and winter, space rather than season explained more of the spatiotemporal variation of soil microbial alpha and beta diversities. Environmental covariates explained some of microbial spatio-temporal variation observed, with fast-changing environmental covariates—climate variables, soil moisture, and available nutrient—likely being the main factors that drove the seasonal variation found in bacterial and fungal beta diversities. Using random forest modeling, we further identified a group of microbial exact sequence variants (ESVs) as indicators of summer and winter seasons and for which relative abundance was associated with fast-changing environmental variables (e.g., soil moisture and dissolved organic nitrogen). Together, our empirical field study's results suggest soil microbial seasonal variation could arise from the changes of fast-changing environmental variables, thus providing integral support to the large emerging body of snapshot studies related to microbial biogeography.

**IMPORTANCE** Both space and time are key factors that regulate microbial community, but microbial temporal variation is often ignored at a large spatial scale. In this study, we compared spatial and seasonal effects on bacterial and fungal diversity variation across an 878-km transect and found direct evidence that space is far more important than season in regulating the soil microbial community. Partitioning the effect of season, space and environmental variables on microbial community, we further found that fast-changing environmental factors contributed to microbial temporal variation.

**KEYWORDS** bacterial community, fungal community, space, season, spatiotemporal variation

Microbial communities regulate vital soil processes, such as nutrient cycling, decomposition rates, and pathogenesis, all of which are fundamental for plant productivity in croplands and natural ecosystems (1, 2). The distribution and ecological drivers of bacteria and fungi are well studied (3) from local to global scales (4, 5). Much less is known, however, of the temporal variability of microbial communities at large spatial scales, as most previous large-scale studies have overlooked potential effects of time in their experimental design, often using samples collected at a single point in

This article followed an open peer review process. The review history can be read here.

Address correspondence to Haiyan Chu, hychu@issas.ac.cn.

time (6–8). Consequently, the relative importance of spatial heterogeneity versus time for driving microbial community composition remains largely unexplored at large spatial scales.

Striking seasonal variation in the soil microbial community and processes at a single or several sampling locations has been well documented for decades, and patterns are commonly attributed to key regulatory factors, such as moisture, temperature, nutrient content, and plant carbon allocation (9–13). Previous studies have provided critical knowledge of major temporal patterns and ecological drivers of local microbial seasonality (14, 15). Yet, much less is known about the relative importance of seasonality and spatial heterogeneity in predicting the distribution of soil microbial communities. Recent meta-data surveys that synthesized spatial and temporal studies from different sites to better understand the biogeographic distribution of microbial communities have highlighted temporal variation as an implication of spatial variation, suggesting both are crucial aspects of microbial biogeography (16, 17). But those meta-analysis studies collected data sets in which either the effect of spatial variation or temporal variation was examined, which could cause site-specific confounding factors, thus obscuring the result interpretation. Hence, it is essential to focus on both spatial and temporal dynamics at the same study sites when trying to disentangle drivers of microbial distributions.

Here, we hypothesized that spatial heterogeneity would be much more important for controlling the alpha and beta diversities of soil microbes than the effect of contrasting seasons (i.e., winter versus summer). In other words, large differences in soil properties across plots are stronger predictors for the distribution of microbial communities than the changes in seasonality they experienced. The reasoning here is that soil microbial communities such as bacteria and fungi are well known to be driven by slow-changing soil properties, such as pH and total organic carbon across large spatial scales (3, 4, 18). These soil properties take from centuries to millions of years to change (19), making them very stable with respect to time. Of course, seasonality could still predict a reduced portion of the variation in microbial communities' distributions that is closely associated with climate factors; in this way, fast-changing soil attributes, such as soil moisture, soil processes, and nutrient pools, which are known to change over days or weeks, are likely better able to predict changes in microbial communities over shorter time periods (days to months). For example, soil moisture is well known to influence the community composition and activity of microbial communities in drying and rewetting processes (20, 21).

To test our hypothesis, we collected soil samples across 45 locations along a ca. 878-km transect during two contrasting seasons (winter versus summer) and used amplicon sequencing to measure soil bacterial and fungal communities. In this way, a total of 90 topsoil samples were collected in wheat croplands across North China Plain. There are two good reasons for choosing wheat fields to investigate spatiotemporal distributions of soil-dwelling microbes. (i) Wheat is among the most economically and functionally important crops globally; therefore, more information on its associated microbial communities is of paramount importance. (ii) By focusing on single plant species (wheat), we could remove the noise derived from having different plant species, which are known to be important drivers of microbial communities at large spatial scales (22).

## RESULTS

**Spatiotemporal distribution of environmental variables.** Among the 12 measured environmental variables, slow-changing environmental factors such as pH, total phosphorus (TP), and total potassium (TK) were highly localized: 87% ($P < 0.01$), 62.6% ($P < 0.01$), and 64.5% ($P < 0.01$) of their respective variation was explained by sampling sites alone. In contrast, fast-changing environmental factors, such as soil moisture (SM; $R^2 = 0.237$, $P < 0.01$), dissolved organic nitrogen (DON; $R^2 = 0.249$, $P < 0.01$), average monthly temperature (Tem; $R^2 = 0.962$, $P < 0.01$), and average monthly precipitation (Pre, $R^2 = 0.352$, $P < 0.01$), were largely influenced by the seasons. In winter, SM and

DON were significantly higher, yet Tem and Pre were significantly lower, than summer (see Fig. S2 in the supplemental material). All the environmental variables were found to be significantly affected by the sampling sites, and environmental dissimilarity significantly increased with geographic distance in both winter and summer (see Fig. S3).

**Spatiotemporal distribution of the soil microbial community.** For the Illumina MiSeq sequencing, 2,053,526 high-quality sequences with 12,381 exact sequence variants (ESVs) and 4,534,055 high-quality sequences with 3,308 ESVs were obtained for the 16S rRNA V4 region and ITS2 region, respectively. At a 99% taxonomy identity threshold, most sequences of 16S rRNA were mainly assigned to *Actinobacteria* (22.19%), *Acidobacteria* (14.07%), and *Proteobacteria* (39.44%) at the phylum level (see Fig. S4a), and ITS2 sequences were mainly assigned to *Dothideomycetes* (17.04%), *Sordariomycetes* (48.37%), and *Agaricomycetes* (7.85%) at the class level (Fig. S4b).

For microbial alpha diversity, season explained 25.2% and 17.2% of the variation, and site explained 38.4% and 23.1% of the variation in soil bacteria and fungi, respectively (Fig. 1a and b). Even though the alpha diversity (the number of ESVs) of bacteria at ZhaoXian (ZX) and PingDu (PD) sites were similar between winter and summer and that of fungi at ShangCai (SC) was significantly higher in winter than in summer, the overall trend in bacterial and fungal alpha diversities was one of being significantly lower in winter than in summer (see Fig. S5). When considering microbial beta diversity (changes in species composition between sites) based on the abundance-related Bray-Curtis distance, we found that site explained 39.6% of bacterial variance and 36.1% of fungal variance, while season only explained 6.1% of bacterial variance and 6.7% of fungal variance (Fig. 1c and d). For microbial beta diversity based on the presence/absence-related Jaccard distance, we also found a much greater site effect than season effect, in which 29.5% of bacterial variance and 27.5% of fungal variance were explained by site, while only 4.8% of bacterial variance and 4.9% of fungal variance were explained by season (see Fig. S6). Despite the soil microbial community showing separation by winter and summer at each site (see Fig. S7 and S8), these divergences were swamped by the impact of sampling sites. In all, we found a stronger geographic location effect than seasonal effect upon both microbial alpha and beta diversity estimates. Exploring the relationship between microbial community similarity and geographic distance, we detected significant distance-decay relationships for bacteria and fungi in both winter and summer. Surprisingly, both bacterial and fungal communities showed higher turnover rates in winter than in summer (Fig. 2a and c). The slight decline in microbial similarity with environmental distance (Euclidean distance between sites based on the measured environmental variables) (Fig. 2b and d) indicated that those environmental variables covarying with season and space could explain part of the observed microbial spatiotemporal variation along the large-scale transect.

**Role of environmental variables in shaping microbial spatiotemporal distributions.** According to the stepwise multiple regression models, environmental variables were able to explain 57% and 31% of the variation in bacterial and fungal alpha diversities, respectively (see Table S1). The environmental variables pH, available potassium (AK), total nitrogen (TN), TK, and Tem were the main factors which induced the shift in bacterial alpha diversity, while DON, available phosphorus (AP), and Tem were those that most changed fungal alpha diversity. Linking microbial beta diversity with environmental variables, we found that pH, which is a highly localized variable, was the main factor involved in shaping both bacterial and fungal beta diversities (see Table S2).

Given that environmental variables changed across space and between seasons, using variation partitioning models let us tease apart the independent effects of season, spatial distance, and environmental variables upon the spatiotemporal dynamics of microbial beta diversity. For the bacterial community, we found that spatial distance, season, and environmental variables together explained 32% of its variations (Fig. 3). The effect of pure space explained 5% of bacterial spatiotemporal variation, whereas

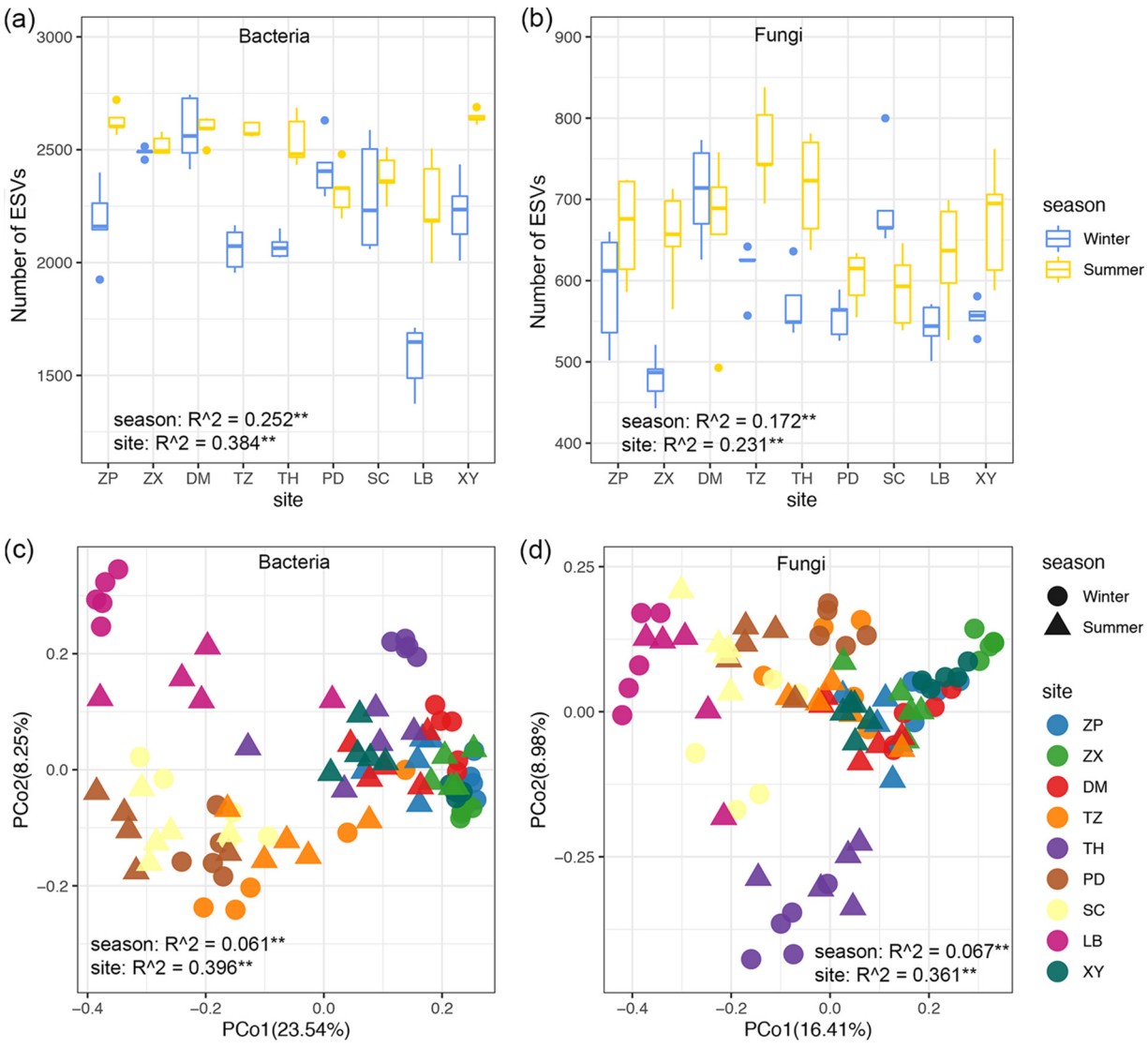

**FIG 1** Boxplots showing the number of ESVs for bacteria (a) and fungi (b) in winter and summer among the 9 sampling sites in China. The significant effect of seasons and site on microbial alpha diversity was detected by two-way ANOVA. The principal-coordinate analysis (PCoA) plots were based on Bray-Curtis dissimilarity of bacterial (c) and fungal (d) communities in winter and summer. The significant effect of seasons and sites on microbial beta diversity was detected by PERMANOVA. **, $P < 0.01$; DM, DaMing; LB, LinBa; PD, PingDu; SC, ShangCai; TH, TaiHe; TZ, TengZhou; XY, XingYang; ZP, ZouPing; ZX, ZhaoXian.

that of pure season did not influence the bacterial spatiotemporal distribution; when combined with fast-changing environmental factors, season did explain 5% of this variation. For the fungal community, 9% of its variation was explained by the effect of pure space, while that of pure season explained just 1% of fungal spatiotemporal dynamics. The fast-changing environmental factors were the main driver of fungal seasonal variation in that they explained 6% of it (Fig. 3).

Partial correlations via redundancy analysis (Table 1) showed that season explained 2.2% of bacterial variation ($P = 0.001$) and 2.8% of fungal variation ($P = 0.001$) when holding the space effect constant. However, when controlling the season effect, space accounted for 15.5% of the bacterial variation ($P = 0.001$) and 16.8% of the fungal variation ($P = 0.001$). Because environmental factors shift across space and over time, they are likely the drivers for the microbial spatial and temporal variation in this study's data set. When controlling the effect of fast-changing environmental variables, season explained a negligible 0.6% of bacterial variation ($P = 0.012$) and 0.7% of fungal variation ($P = 0.004$), whereas space explained 11.6% of this bacterial variation

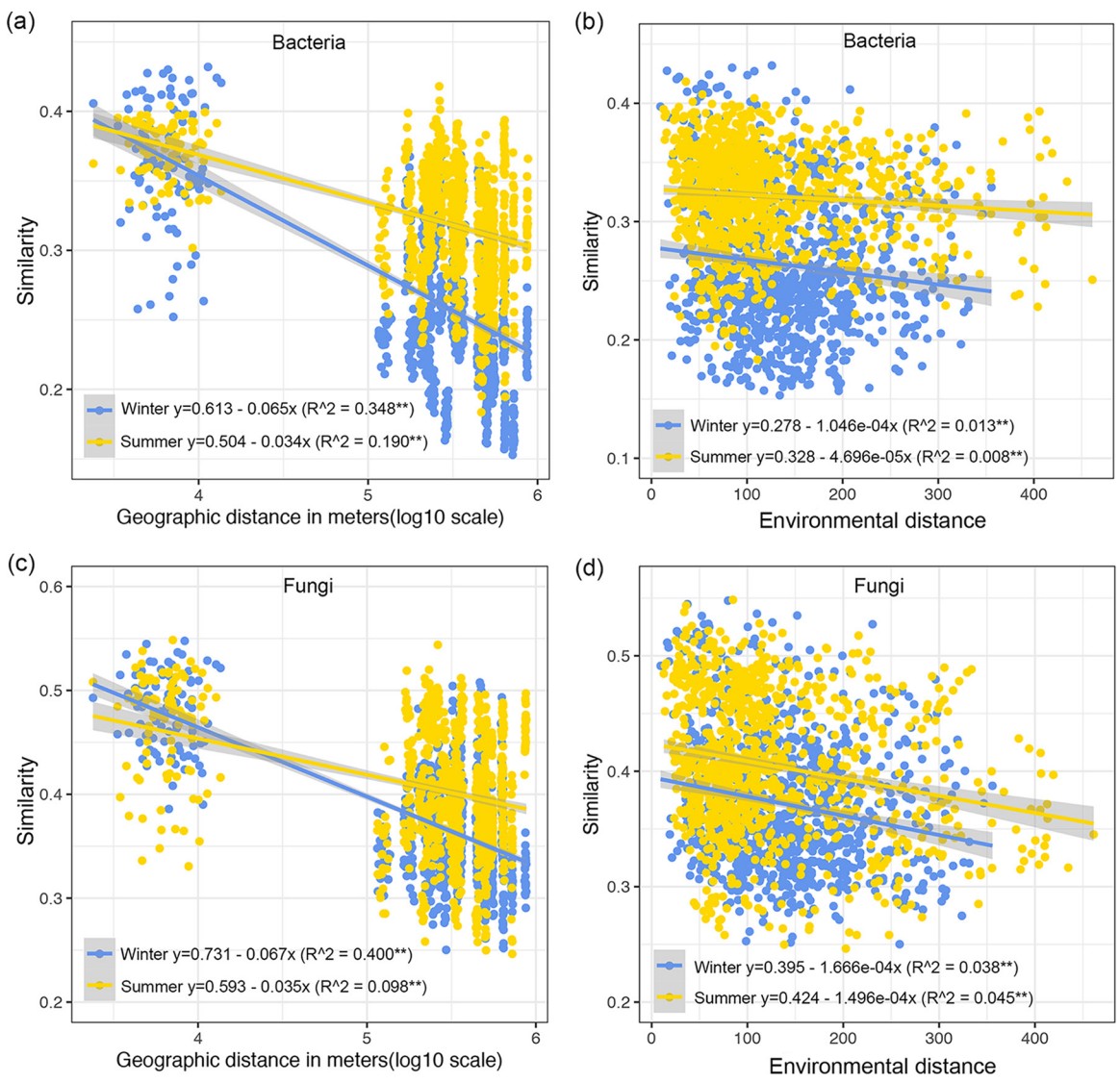

**FIG 2** Distance-decay relationships of bacterial (a) and fungal (c) communities (based on Jaccard distances) in winter and summer, and the linear relationships between bacterial (b) and fungal (d) similarities with environment distance (based on Euclidean distance). **, $P <$ 0.01.

($P = 0.001$) and 12.8% of this fungal variation ($P = 0.001$). After controlling for the effect of slow-changing environmental attributes, season explained 2.2% of bacterial variation ($P = 0.001$) and 2.5% of fungal variation ($P = 0.001$), and space explained 11.1% of bacterial variation ($P = 0.001$) and 13.1% of fungal variation ($P = 0.001$). The sharply diminished seasonal effect on microbial spatiotemporal distribution when controlling the contribution of fast-changing environmental factors revealed that the fast-changing factors figure prominently in determining microbial seasonal variation.

Finally, a random forest model was used to distinguish those microbial ESVs which could discriminate microbial community in the winter and summer. Ranked by their importance value, the top 20 bacterial ESVs mainly belonged to *Gammaproteobacteria* and *Alphaproteobacteria* (Fig. 4a), while the top 20 fungal ESVs mainly belonged to *Sordariomycetes* (Fig. 4b). The relative abundance of those ESVs was strongly correlated with the fast-changing environmental properties SM, DON, Tem, and Pre (Fig. 4). This result also implied that seasonal variations in soil bacterial and fungal communities were mainly induced by fast-changing environmental variables.

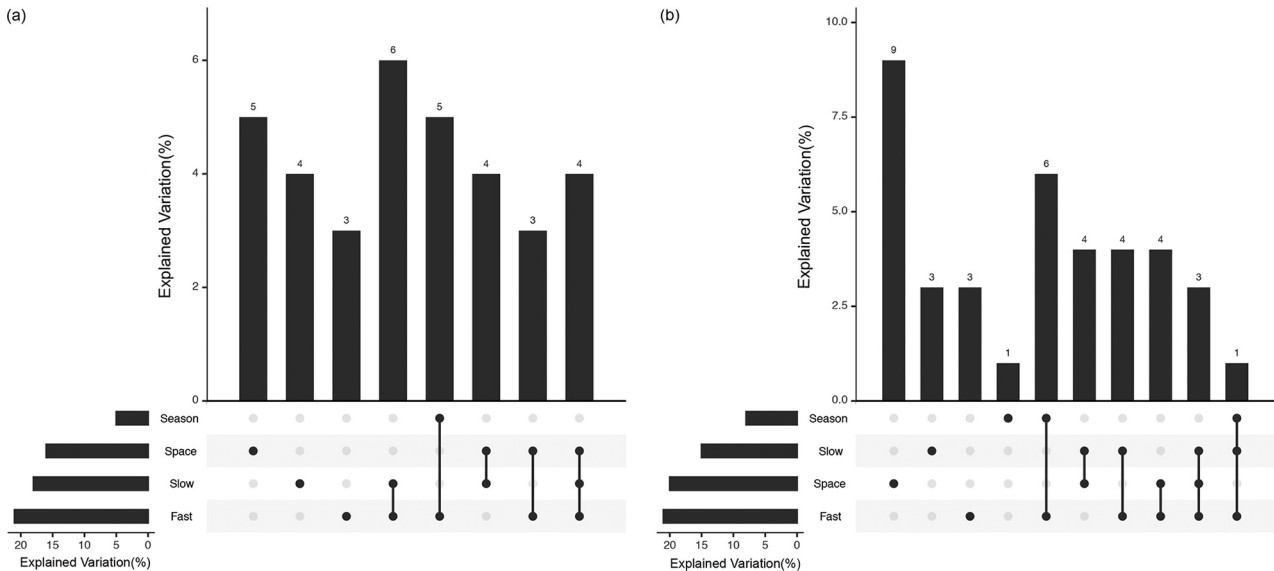

**FIG 3** UpSet plot showing the results from variation partitioning models that were used to identify the effects of season, space, and fast and slow environmental variables. Fast, fast-changing environmental variables were soil moisture, dissolved organic nitrogen, available phosphorus, available potassium, dissolved organic carbon, average monthly temperature, and average monthly precipitation; Slow, slow-changing environmental variables consisted of pH, organic carbon, total nitrogen, total phosphorus, and total potassium.

## DISCUSSION

Many studies investigating soil microbial communities through time at the local scale have revealed large temporal variability in their structures (23, 24), while the unexplained variation of large-scale biogeographic studies sampled over months or

**TABLE 1** Partial correlations between bacterial or fungal beta diversity and season or space in combination with fast and slow after controlling for each other's effect by redundancy analysis[a]

| Variable(s) | Bacteria | | Fungi | |
|---|---|---|---|---|
| | Variance | P value | Variance | P value |
| Control space | | | | |
| Season | 0.022 | 0.001 | 0.028 | 0.001 |
| Season+fast | 0.075 | 0.001 | 0.075 | 0.001 |
| Season+slow | 0.072 | 0.001 | 0.067 | 0.001 |
| Season+fast+slow | 0.103 | 0.001 | 0.099 | 0.001 |
| | | | | |
| Control season | | | | |
| Space | 0.155 | 0.001 | 0.168 | 0.001 |
| Space+fast | 0.209 | 0.001 | 0.215 | 0.001 |
| Space+slow | 0.205 | 0.001 | 0.208 | 0.001 |
| Space+fast+slow | 0.237 | 0.001 | 0.24 | 0.001 |
| | | | | |
| Control fast | | | | |
| Season | 0.006 | 0.012 | 0.007 | 0.004 |
| Season+slow | 0.056 | 0.001 | 0.051 | 0.001 |
| Space | 0.116 | 0.001 | 0.128 | 0.001 |
| Space+slow | 0.145 | 0.001 | 0.153 | 0.001 |
| | | | | |
| Control slow | | | | |
| Season | 0.022 | 0.001 | 0.025 | 0.001 |
| Season+fast | 0.069 | 0.001 | 0.079 | 0.001 |
| Space | 0.111 | 0.001 | 0.131 | 0.001 |
| Space+fast | 0.158 | 0.001 | 0.181 | 0.001 |

[a]Permutation testing was used to assess the significance of the constraints. Fast, fast-changing environmental variables, namely, soil moisture, dissolved organic nitrogen, available phosphorus, available potassium, dissolved organic carbon, average monthly temperature, and average monthly precipitation; Slow, slow-changing environmental variables, namely, pH, organic carbon, total nitrogen, total phosphorus, and total potassium.

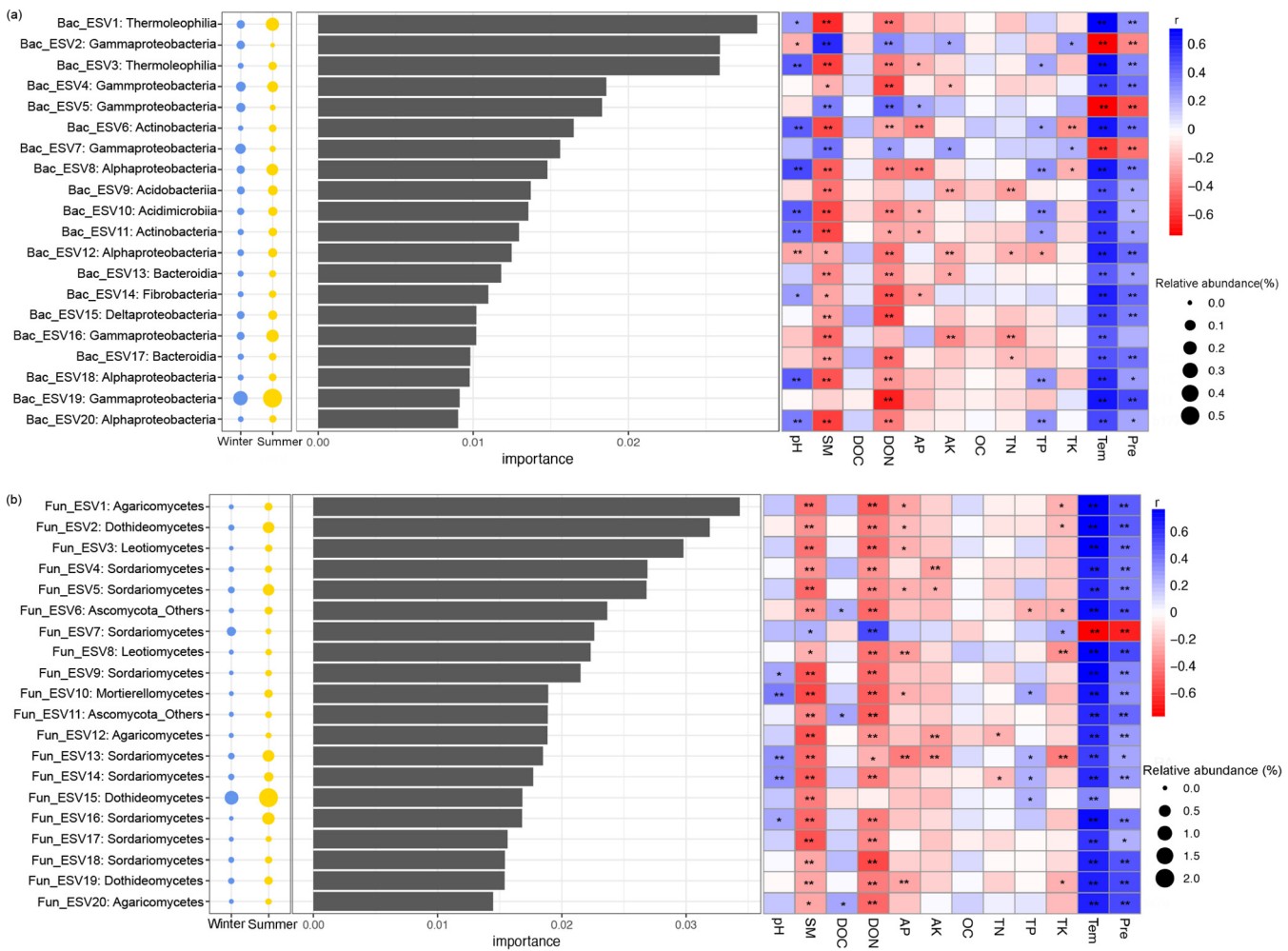

**FIG 4** Top 20 identity ESVs used for discriminating bacterial (a) and fungal (b) communities in winter and summer (detected by random forest model). The assigned taxonomy of each taxon is displayed at the class level. The bubbles on the left show the relative abundances of those ESVs in winter and summer; the middle bar plots show the importance value of each ESV estimated by random forest model; the right heat maps show the Spearman correlations between the relative abundances of identity ESVs and environmental variables. SM, soil moisture; DOC, dissolved organic carbon; DON, dissolved organic nitrogen; AP, available phosphorus; AK, available potassium; TN, total nitrogen; TP, total phosphorus; TK, total potassium; Tem, average monthly temperature; Pre, average monthly precipitation; *, $P < 0.05$; **, $P < 0.01$.

years (4, 5, 7) can be attributed to temporal variance. However, few large-spatial-scale studies of soil microbial biogeography actually sample the same locations at multiple times. In this study, we investigated the spatiotemporal patterns of soil bacterial and fungal communities along ca. 878-km in two contrasting seasons, because doing so can help to evaluate the independent effects of time and space upon microbial community variations. Our results provide solid empirical evidence that spatial heterogeneity is, by far, more important than seasonality for predicting the spatiotemporal variation that characterizes the microbial alpha and beta diversity of the studied landscape. Crucially, our work showed that seasonal variation in microbial community composition was driven by fast-changing environmental factors.

Although a few studies did point out that sampling time is less important than sampling space on microbial community distributions over a large spatial scale (25, 26), since microbial communities can differ even on the scale of meters or even centimeters (27), no study to date provided the direct evidence required to compare the effects of temporal and spatial dynamics on soil microbial community distributions on a large spatial scale. Here, we provided field evidence that demonstrates spatial variability exceeded seasonal variability on both bacterial and fungal spatiotemporal distributions when assessed at a large spatial scale. However, the temporal signals could have been

masked by the presence of relic DNA (DNA released from dead microbes) in our data. For example, in a study designed to disentangle the relationships of spatial, temporal, and relic DNA effects on soil microbial communities on opposite hillslopes in Colorado (USA), removing relic DNA from soil led to greater temporal variations (13). In our study, total bacterial and fungal DNA was investigated without considering the effect of relic DNA, which might underestimate the true extent of temporal variation in microbial community dynamics.

We also uncovered evidence indicating that fast-changing environmental factors underpin the mechanisms responsible for soil microbial seasonal variation in North China Plain. Many previous studies that focused on seasonal variation in microbial composition implicate that it is often induced by availability of nutrients, temperature, and moisture (28, 29). However, not all taxa within the microbial community are equally sensitive to temporal changes in the environment (30). In our study, we detected certain taxa that contributed to the discrimination of winter and summer microbial communities, most of them being *Alphaproteobacteria*, *Gammaproteobacteria*, and *Sordariomycetes*, all of which had strong relationships to the fast-changing environmental variables of soil moisture, dissolved organic nitrogen, temperature, and precipitation. Those temporally sensitive microbial taxa also support the findings that microbial temporal variations were regulated by fast-changing environmental variables. According to their life strategies, the *Alphaproteobacteria* and *Gammaproteobacteria* have been assigned as copiotrophic bacteria (31). So, it is not surprising to find that those copiotrophic bacteria responded to altered soil moisture, temperature, and precipitation conditions, because their sensitivity to the first of these factors in particular has already been demonstrated (32, 33). Compared with microbial temporal variation, microbial spatial variation is often affected by pH, an endemic soil property, in both nature and agricultural ecosystems (34–36). Covariate environmental factors likely were not the main mechanisms shaping microbial spatial variation, as we found that controlling for their effects only marginally reduced the spatial effect. More than 60% of the variations in microbial composition were not explained by season, spatial distance, or environmental variables in our study. The possible reason for this result might be the existence of other unmeasured environmental factors that vary in space and time (16, 17), including biotic interactions such as competition, mutualism, and predation between microbial taxa (37, 38) and ecological processes such as dormancy and persistence traits of microbial communities and their members (8).

Nonetheless, the bacterial and fungal communities both attained higher species richness in the summer; they showed lower spatial turnover than in winter. Microbial taxonomic richness is known to reflect metabolic diversity (39). Thus, the greater species diversity in summer probably arose from high enzyme activity promoted by favorable temperatures and light conditions characteristic of that season (40). The higher similarity in community compositions across locations in summer versus winter suggests that despite the higher richness of soil microbes in this season, many of these microbes could be found across different locations, thereby reducing the beta diversity of these organisms at the regional scale. Even though fungi and bacteria displayed similar degrees of spatiotemporal variation in this study, the respective influence of environment and spatial distance upon bacterial and fungal alpha and beta diversities was distinct. Environmental variables were able to explain nearly twice the variation in bacterial than in fungal alpha diversity; this might suggest that fungal communities are less responsive to seasonal changes than bacteria. However, spatial distance exerted a larger effect on the beta diversity of fungi than on that of bacteria. The weaker environmental effect on fungal alpha diversity coupled to the stronger spatial effect on fungal beta diversity could be linked and tied to the fact that bacteria are more apt to be affected by local changes in soil properties (25), while the evolved life history of fungi (hyphae formations and viable durable spores) make them more tolerant of sudden environmental changes (41). Moreover, the generally greater individual body size of fungal than bacterial members of the community would entail more severe dispersal limitations (42, 43). This could result in larger spatial distance effect as well as

the priority effect—those microorganisms first arriving at a site for colonization can significantly affect the establishment of later arriving species—which would have greater influence on the fungal community than on the bacterial community (41, 44), thus contributing to the larger spatial effect we found in China.

Although the soil microbial community showed clear seasonal variability within sampling sites, the effect of spatial heterogeneity was far more important than season in regulating the compositions of both bacteria and fungi at the large spatial scale. Fast-changing environmental factors affected by time were thus contributing to the mechanisms driving microbial temporal variation. These results indicate that, to some extent, microbial distribution patterns at a large spatial scale can be roughly predicted by using data obtained in snapshot studies, since the temporal variation may be explained by environmental factors. But caution should be taken when interpreting our results, given that only two time points, summer and winter, were included in our study. In the temperate zone, plant productivity peaks in the summer because of the favorable temperature and light conditions, whereas photosynthate inputs are considered negligible in winter when it is cold and there is too little light (15). Accordingly, examining only those two seasons would probably lead to the most pronounced environmental difference in natural ecosystems. However, crop management practice time points (e.g., fertilization, planting, and harvest of wheat) could also have induced significant changes in microbial temporal variation (45, 46) in the wheat field landscape studied here. The limited time points used in this study might thus underestimate true microbial temporal variation; hence, more time points should be included when designing future similar studies (including those of microbial biogeography). To conclude, investigations of soil microbial ecology in both space and time are those which are most likely to provide a richer comprehensive understanding of the key factors regulating the biodiversity of the soil ecosystems.

## MATERIALS AND METHODS

**Soil sampling.** Our sampling sites were located in North China Plain, where winter wheat cultivation has been ongoing for more than 40 years (36). To explore microbial spatiotemporal distributions during the winter wheat growing season, 45 topsoil (0 to 15 cm) samples were collected from 9 sites at each time point—in November 2014 after irrigation and fertilization, and again in May 2015, nearly 3 weeks before harvest—for a total of 90 samples. The 9 sites located on North China Plain spanned ca. 878 km (see Fig. S1 in the supplemental material). Since its soils are highly heterogeneous, 5 plots (4 in corners and 1 in center) were set up for collecting soil samples within a 100-km$^2$ quadrat in each site, with any two plots at least 5 km apart. To collect representative soil samples per plot, a cost-effective and optimal heterogenous site sampling method compositing (47) was used. Specifically, 12 cores were collected by drill (3-cm diameter) in each plot and then mixed together to form a composite single sample (Fig. S1). All these samples were then delivered on ice to the lab within 3 days. There, they were sieved through a 2-mm mesh and divided into two subsamples: one stored at 4°C for the analysis of soil physical and chemical properties and the other stored at $-20$°C for the DNA extractions. Latitude and longitude information were collected by global positioning system (GPS) when sampling. Corresponding average monthly temperature (Tem) data in November 2014 and May 2015 were downloaded from MOD11C3 in Modis (https://terra.nasa.gov/about/terra-instruments/modis), and likewise, the average monthly precipitation (Pre) data for both times were acquired from the GPM_3IMERGM database (https://disc.gsfc.nasa.gov/datasets/GPM_3IMERGM_06/summary).

**Soil physical and chemical property analyses.** Soil pH was measured with a pH meter (Thermo Orion-868; Thermo Fisher Scientific, MA, USA) in a 1:5 soil-to-water ratio. Soil moisture (SM) was measured by the gravimetrical method. Dissolved organic carbon (DOC) extracted by deionized water and dissolved organic nitrogen (DON) extracted by 0.5 M $K_2SO_4$ were determined by an organic carbon-nitrogen analyzer (Shimadzu, Kyoto, Japan). Available phosphorus (AP) extracted by 0.5 M $NaHCO_3$ and total phosphorus (TP) were measured by the molybdenum blue method, while available potassium (AK) extracted by 1 M $CH_3COONH_4$ and total potassium (TK) were measured using the flame photometry method. Organic carbon (OC) was determined by applying a traditional dichromate oxidation titration. Total nitrogen (TN) was measured via combustion. All of these measurements were according to the instructions described in our previous study of the biogeographical distribution of bacterial communities in wheat fields (36).

**Microbial community analysis.** Soil nucleic acids of all the samples ($n = 90$) were extracted using the FastDNA Spin kit for soil (MP Biomedicals, Santa Ana, CA) and purified by the Ultra Clean 15 DNA purification kit (MO BIO, Carlsbad, CA, USA). Next, the DNA concentration of each sample was quantified in a NanoDrop ND-1000 spectrophotometer (Thermo Scientific, Wilmington, Germany), after which it was stored at $-20$°C for later sequencing.

The primer pair 515F (5'-GTGCCAGCMGCCGCGGTAA-3') and 806R (5'-GGACTACHVGGGTWTCTAAT-3') was used to amplify the 16S V4 hypervariable region (48), and likewise, the ITS3 (5'-GCATCGATGA AGAACGCAGC-3') and ITS4 (5'-TCCTCCGCTTATTGATATGC-3') primer pair was used to amplify the fungal ITS2 region (49). Each sample was amplified in a 30-$\mu l$ reaction mixture with 15 $\mu l$ Phusion high-fidelity PCR master mix (New England BioLabs), 0.2 $\mu l$ each of forward and reverse primers, and ca. 10 ng of template DNA. The reaction conditions for the 16S V4 region consisted of 30 cycles of denaturation at 94°C for 30 s, annealing at 55°C for 30 s, and extension at 72°C for 30 s; those for the ITS2 region consisted of 30 cycles of denaturation at 98°C for 10 s, annealing at 50°C for 30 s, followed by extension at 72°C for 30 s. All the PCR products were purified by a QIAquick PCR purification kit (Germany Qiagen) and quantified in the NanoDrop ND-1000 spectrophotometer. These purified PCR products was amassed together for Illumina MiSeq sequencing.

The paired-end raw reads were merged by FLASH (50), and ensuing merged reads were then assigned to each sample based on unique barcodes. The QIIME2 2018.08 pipeline was used for sequence quality control and to estimate their diversity. The Deblur algorithm was used, at single-nucleotide resolution, to reduce the inherent noise in the PCR and DNA sequencing (51). According to the merged sequence quality, all sequences were trimmed to 280 bp for 16S and 180 bp for ITS, to avoid introducing study-specific biases. This resulted in exact sequences variants (ESVs) having a high resolution exceeding the 97% identity threshold for operational taxonomic units (OTUs). After Deblur denoising, a *de novo* chimera filtering method, applied with vsearch, was used to remove any chimeras. Taxonomic classification of the ESVs was carried out by applying the prefitted sklearn-based taxonomy classifier to the Silva database (132 release) for 16S, and likewise to the UNITE database (17-12 release) for ITS at a 99% shared identity. Singletons were filtered out, and 9,127 sequences for 16S and 14,313 sequences for ITS were randomly selected to rarify the data sets to the same sampling effort for alpha and beta diversity comparisons.

**Statistical analysis.** The relative importance of the nine sites and two seasons for explaining the variation in environmental variables and the number of microbial (bacteria and fungi) ESVs were evaluated by two-way analysis of variance (ANOVA). The permutational multivariate analysis of variance (PERMANOVA) in the R vegan package was used to test the variations in microbial community structure as explained by sites and seasons. Relationships between the environmental variables and the number of microbial ESVs were calculated by stepwise multiple regression models, while the contribution of environmental variables to microbial community structure based on Bray-Curtis (relative abundance) and Jaccard (presence/absence) distance matrices were assessed by PERMANOVA. Microbial distance-decay relationships were estimated between microbial Jaccard dissimilarity and geographic distance.

Variation partitioning (52) was used to quantify the relative importance of four groups of predictors, namely, fast-changing environmental properties (SM, DON, DOC, AP, AK, Tem, and Pre), slow-changing environmental factors (pH, OC, TN, TK, and TP), season, and space, on the variation observed in soil microbial community composition. Latitudinal and longitudinal site data of each site were transferred to rectangular data to represent spatial distance by function pcnm(), and variation partitioning analyses were conducted with function varpart() in the vegan package for R. All the environmental variables were standardized by transforming their values to Z-scores, so as to remove the unit difference of each variable. Visualization of the variation partitioning result was performed by function upset() in the UpSetR package for R. Environmental distance was calculated as the Euclidean distance between sites based on all the measured environmental variables. Partial correlations were performed by rda() in the R vegan package to estimate one factor's influence upon microbial variation when controlling the other ones. To find the best discriminant microbial ESVs in the two seasons, classification random forest analysis was applied by using sklearn module in Python v3.6. Spearman correlations were used to estimate the relationships between those identity ESVs and the environmental variables. Plots were drawn by ggplot2 in R 3.4.3.

**Data availability.** The raw sequencing data for bacterial and fungal communities have been submitted to the National Center of Biotechnology Information (NCBI) Sequence Read Archive under accession number PRJNA508409.

## SUPPLEMENTAL MATERIAL

Supplemental material is available online only.

**FIG S1**, TIF file, 0.7 MB.
**FIG S2**, TIF file, 2.0 MB.
**FIG S3**, TIF file, 0.8 MB.
**FIG S4**, TIF file, 1.4 MB.
**FIG S5**, TIF file, 0.6 MB.
**FIG S6**, TIF file, 1.8 MB.
**FIG S7**, TIF file, 1.2 MB.
**FIG S8**, TIF file, 1.1 MB.
**TABLE S1**, XLSX file, 0.1 MB.
**TABLE S2**, XLSX file, 0.1 MB.

## ACKNOWLEDGMENTS

We thank Yu Shi, Yuntao Li, Teng Yang, Xingjia Xiang, Congcong Shen, Ruibo Sun, Dan He, Yingying Ni, and Kunkun Fan for their assistance in soil sampling and laboratory analyses and also Liang Chen and Jie Liu for collecting the climate data.

This work was supported by the Strategic Priority Research Program of the Chinese Academy of Sciences (XDB15010101) and the China Biodiversity Observation Networks (Sino BON). M.D-B. is supported by the Spanish Government under a Ramón y Cajal contract (RYC2018-025483-I) and by a large research grant from the British Ecological Society (grant agreement no. LRA17\1193, MUSGONET).

We have no conflict of interest to declare.

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
