## [Reviewer comments · mSystems]

Space is more important than season when shaping soil microbial communities at large spatial scale

Kaoping Zhang, Manuel Delgado-Baquerizo, Yong-Guan Zhu, and Haiyan Chu

Corresponding Author(s): Haiyan Chu, Institute of Soil Science, Chinese Academy of Sciences

Review Timeline:

Submission Date:	November 19, 2019
Editorial Decision:	February 14, 2020
Revision Received:	March 25, 2020
Editorial Decision:	April 17, 2020
Revision Received:	April 21, 2020
Accepted:	April 22, 2020

Editor: Marnix Medema

Reviewer(s): The reviewers have opted to remain anonymous.

Transaction Report:

DOI: <https://doi.org/10.1128/mSystems.00783-19>

February 14, 2020

Prof. Haiyan Chu
Institute of Soil Science, Chinese Academy of Sciences
71 East Beijing Road
Nanjing 210008
China

Re: mSystems00783-19 (Spatial changes are far more important than seasonality when predicting the large-scale distribution of soil microbes)

Dear Prof. Haiyan Chu:

Thank you for submitting your manuscript to mSystems.

As you can see, the two reviewers found merit in some aspects of your work but also raised substantial concerns. With regard to the replicates, I think it is currently insufficiently clear to what extent the 5 samples per county (for 9 counties) serve as de facto replicates (see comments from reviewer 1). This needs to be clarified in the text. Contextualization with other literature should also be improved, as indicated by the reviewers.

Below you will find the comments of the reviewers.

To submit your modified manuscript, log onto the eJP submission site at <https://msystems.msubmit.net/cgi-bin/main.plex>. If you cannot remember your password, click the "Can't remember your password?" link and follow the instructions on the screen. Go to Author Tasks and click the appropriate manuscript title to begin the resubmission process. The information that you entered when you first submitted the paper will be displayed. Please update the information as necessary. Provide (1) point-by-point responses to the issues raised by the reviewers as file type "Response to Reviewers," not in your cover letter, and (2) a PDF file that indicates the changes from the original submission (by highlighting or underlining the changes) as file type "Marked Up Manuscript - For Review Only."

Please return the manuscript within 60 days; if you cannot complete the modification within this time period, please contact me. If you do not wish to modify the manuscript and prefer to submit it to another journal, please notify me of your decision immediately so that the manuscript may be formally withdrawn from consideration by mSystems.

To avoid unnecessary delay in publication should your modified manuscript be accepted, it is important that all elements you upload meet the technical requirements for production. I strongly recommend that you check your digital images using the Rapid Inspector tool at <http://rapidinspector.cadmus.com/RapidInspector/zmw/>.

Please also make sure to upload your data (FASTQ files, etc.) to public repositories, and indicate accession numbers in a 'Data Availability' section.

If your manuscript is accepted for publication, you will be contacted separately about payment

when the proofs are issued; please follow the instructions in that e-mail. Arrangements for payment must be made before your article is published. For a complete list of **Publication Fees**, including supplemental material costs, please visit our website.

Sincerely,

Marnix Medema

Editor, mSystems

Journals Department
Reviewer comments:

Reviewer #1 (Comments for the Author):

The whole text suffers from very poorly-used language, grammatical mistakes and typos. This makes understanding of the text highly difficult, if not impossible. I strongly suggest the re-writing of the text.

I appreciate collection of the soil samples from wide number of sites (45 sites from two seasons). However, lack of replicates from each collection site makes interpretation of the results unreliable. For example, the study does not consider heterogeneity of soil communities within a location. I suggest authors to discuss this issue in the discussion.

The main conclusion of the study contradicts with the previous literature on spatial and temporal changes in soil microbiome (e.g. Noah Fierer, 2017, Nature Plants). I recommend the authors to discuss this in the discussion.

Reviewer #2 (Comments for the Author):

In this study, variation in microbial community structure is partitioned into space, time, and environmental conditions. Soil samples were collected from 45 locations across a transect of wheat fields in the summer and the winter, and measurements of soil chemistry and climate were

compiled. Alpha diversity (number of ESVs) was modeled via stepwise regression as a function of environmental variables. The authors report that environmental variables explain 31% of variation diversity of fungal communities and 57% of diversity in bacterial communities, with pH having the strongest effect. Community structure (in the form of Bray-Curtis or Jaccard distance) was modeled as a function of spatial distance, season, and environmental variables. 32% of the variation in bacterial community structure was explained by all predictors, but the total is not directly reported for fungi. Season explained very little change in community structure, and the small amount explained for fungi (9% of variation) was mostly explained by environmental variables once they were included.

Some of the terminology was a bit confusing upon my first pass through the paper. For instance, using "ESVs" to refer to the number of unique ESVs per sample: instead, the term "ESV diversity" or "alpha diversity" may be more readily understood. Line 134 mentions "community structure", then on line 139, "spatio-temporal variation" is used, and then "community variations." If these are all referring to alpha diversity or Bray-curtis distance, the terminology should be standardized.

In the discussion, the main conclusions are that spatial variability exceeds temporal variability, and that temporal variability can mostly be explained by environmental conditions. However, this is then taken to mean that snapshot studies are sufficient for explaining patterns of spatial scale, since seasonal effects are captured by environmental variables. This may be an unsupported conclusion, because only two sampling dates / seasons were considered: there is no discussion of the potential for interannual variability, or the possibility that other seasons may have strong patterns (such as decomposers flourishing in the fall, or wheat being planted/harvested, etc.).

I would advocate for more discussion about the significance and limitations of your results, as well as the appropriate next steps. For instance, environmental variables could explain twice the amount of bacterial diversity compared to fungal diversity - why might that be? Do you expect that fungal communities are more affected by priority effects, or soil treatments? The difference in predictability between the two domains is very interesting, but is not treated as substantial within the current manuscript.

The analysis of which microbial taxa differentiate communities (Fig. 4) is only given one sentence in the results, and it is not included in the discussion at all. The significance of this figure should be described further - possibly, are any of these taxa ecologically significant? Do these patterns confirm or refute other trends in the literature?

Line 119 - Were all community structure analyses performed with Bray-Curtis AND Jaccard, and were the conclusions (qualitatively) the same? If so, consider making that more explicit.

Line 191 - This sentence includes undefined terms (such as legacy effect) and is repetitive: "In fact, it's reasonable the contemporary environmental factors only explained part of microbial spatial variation as the habitat type, space isolation and the legacy effect were likely regulate microbial spatial variations (14, 33). The large proportion of variations unexplained by season, spatial distance and environmental variables detected in our study might induced by the unmeasured environmental factors vary in space and time (14-15), biotic interaction (34-35) and ecological processes such as dormancy and persistence traits (6)."

This could be expanded on to discuss what types of biotic interactions may be relevant.

Figure S5(b) - The figure text states that R2 for site is much higher than R2 for season, and that "Even though, the microbial community separated by winter and summer in each site, the

differentiations were masked by sampling sites." I think that merits a bit more explanation. Did this PCoA include environmental variables that could explain the grouping?

Figures 1 and 3 should be more explicit about what the response variables and the predictors are. Figure 3 is a rather confusing way to communicate the variance partitioning. A stacked bar plot may be more interpretable. If you proceed with a 4-way venn diagram, I recommend including more informative labels, and some emphasis on what parts of the diagram the reader is supposed to pay attention to.

Figure 4 lists many duplicate taxa, suggesting a problem with this figure. I know some names, like Actinobacteria, can refer to a phylum as well as a class, but "Sordariomycetes" is repeated 9 times, and Gammaproteobacteria is repeated 6 times, and the legend claims that all are at the class level.

Re-writing of the text:

Understanding of the text is highly difficult, if not impossible. The whole text suffers from very poorly-used language, grammatical mistakes and typos.

1: Title should be reformulated.

51: More examples from the literature are needed.

85: The authors mention abruptly about “wheat” at the end of the text.

90-91: Introduction is not right place to indicate statistical tests used.

Results:

95-103: I am missing any reports about p-vals etc. Too many abbreviations are making the reading of the text difficult.

140-161: Again, statistical tests, p-vals etc. are not reported.

Figures:

Figure 1: Difference in alpha diversity between winter and summer is not the case in some of the sites (e.g. ZX-bacterial communities). Also, higher diversity in samples collected in summer is not always the case in all sites.

Dear Dr. Medema

Thank you very much for the time you have spent on handling our manuscript (mSystems00783-19). We greatly appreciate the opportunity to address all comments from the editor and reviewers. Please find the attached updated manuscript and point-by-point response to the reviewers' comments.

Thank you for your time and attention!

Your sincerely,

Haiyan Chu

For the authors

March 25, 2020

Reviewer #1

The whole test suffers from very poorly-used language, grammatical mistakes and typos. This makes understanding of the text highly difficult, if not impossible. I strongly suggest the re-writing of the text.

Response: We have now gone through the entire text aiming to improve the grammatical quality of our work and remove typos. We have also send our paper to professional grammar revisions to further improve the readability of our manuscript.

I appreciate collection of the soil samples from wide number of sites (45 sites from two seasons). However, lack of replicates from each collection site makes interpretation of the results unreliable. For example, the study does not consider heterogeneity of soil communities within a location. I suggest authors to discuss this issue in the discussion.

Response: We appreciate this comment and apologies for the misunderstanding. In fact, we included replication within each site. In brief, we selected 9 sites located in North China Plain. In each of these sites, we collected 5 replicate samples within a 10km x 10km quadrat. In each quadrat, twelve cores were collected by 3 cm diameter drill and composited as a single sample. We have now clarified this important point in lines 276-286 and have included a supplementary figure S1 with our sampling design.

The main conclusion of the study contradicts with the previous literature on spatial and temporal changes in soil microbiome (e.g. Noah Fierer, 2017, Nature Plants). I recommend the authors to discuss this in the discussion.

Response: Thanks for the suggestion. We checked papers of Noah Fierer's lab published during 2016-2020. Fierer's lab latest paper on the relative importance of space and time in controlling microbial communities suggest that within-plot spatial variability is more important than time within a given location (Carini et al., 2020). Our work goes a step further and suggest that, at the large spatial scale, space is also more

important than time in controlling microbial communities. This result is integral to validate the work being done at the large spatial scale, based on snapshots – as space is expected to be far more important than temporal variability. We have clarified this important point in lines 195-203.

Reference:

Carini P, Delgado-Baquerizo M, Hinckley ES, Holland-Mortiz H, Brewer TE, Rue G, Vanderburgh C, McKnight D, Fierer N. 2020. Effects of spatial variability and relic DNA removal on the detection of temporal dynamics in soil microbial communities. MBio, 11(1), e02776-19.

Reviewer#1 Review attachment 1

Re-writing of the text:

Understanding of the text is highly difficult, if not impossible. The whole text suffers from very poorly-used language, grammatical mistakes and typos.

Response: Thank you. We have now sent our paper to professional company to improve the readability of our work and correct grammatical mistakes.

1: Title should be reformulated

Response: Revised in lines 3-4.

51: More examples from the literature are needed

Response: Revised in line 52.

85: The author mention abruptly about “wheat” at the end of the text.

Response: We reshaped the expression in lines 86-89.

90-91: Introduction is not right place to indicate statistical test used.

Response: We have deleted this description.

Results:

95-103: *I am missing any reports about p-vals etc. Too many abbreviations are making the reading of the text difficult.*

Response: Revised in lines 96-101.

140-161: *Again, statistical tests, p-vals etc. are not reported.*

Response: P-vals were reported in Table 1 by redundancy analysis and have been added in lines 156-166.

Figures:

Figure 1: Difference in alpha diversity between winter and summer is not the case in some of the sites (e.g. ZX-bacterial communities). Also, higher diversity in samples collected in summer is not always the case in all sites.

Response: We agreed that microbial diversity in summer was not always significantly higher than that in winter. The bacterial alpha diversity in ZX and PD didn't show significant change between winter and summer, and fungal diversity in SC was significantly higher in winter than summer, but the total trend of bacterial and fungal alpha diversity was significantly lower in winter and summer (Fig. S5). We have now clarified this important point in lines 116-119.

Reviewer #2

In this study, variation in microbial community structure is partitioned into space, time, and environmental conditions. Soil samples were collected from 45 locations across a transect of wheat fields in summer and the winter, and measurements of soil chemistry and climate are compiled. Alpha diversity (number of ESVs) was modeled via stepwise regression as a function of environmental variables. The authors report that environmental variables explain 31% of variation diversity of fungal communities and 57% of diversity in bacterial communities, with pH having the strongest effect. Community structure (in the form of Bray-Curtis or Jaccard distance) was modeled as a function of spatial distance, season, and environmental variables. 32% of the variation in bacterial community structure was explained by all predictors, but the total is not directly reported for fungi. Season explained very little change in community structure, and the small amount explained for fungi (9% of variation) was mostly explained by environmental variables once they were included.

Some of the terminology was a bit confusing upon my first pass through the paper. For instance, using “ESVs” to refer to the number of unique ESVs per sample: instead, the term “ESV diversity” or “alpha diversity” may be more readily understood. Line 134 mentions “community structure”, then on line 139, “spatio-temporal variation” is used, and then “community variations.” If these are all referring to alpha diversity or Bray-curtis distance, the terminology should be standardized.

Response: Thank you for your positive and constructive comments. We have now updated the terminology in our paper to alpha diversity and beta diversity.

In the discussion, the main conclusions are that spatial variability exceeds temporal variability, and that temporal variability can mostly be explained by environmental conditions. However, this is then taken to mean that snapshot studies are sufficient for explaining patterns of spatial scale, since seasonal effects are captured by environmental variables. This may be an unsupported conclusion, because only two sampling dates/seasons were considered: there is no discussion of the potential for interannual variability, or the possibility that other seasons may have strong patterns (such as decomposers flourishing in the fall, or wheat being planted/harvested, etc.)

Response: Thank you. We have now acknowledged the limitations (only two sampling dates) associated with the conclusions and clarified in lines 259-269.

I would advocate for more discussion about the significance and limitations of your results, as well as the appropriate next steps. For instance, environmental variables could explain twice the amount of bacterial diversity compared to fungal diversity – why might that be? Do you expect that fungal communities are more affected by priority effects, or soil treatments? The difference in predictability between the two domains is very interesting, but is not treated as substantial within the current manuscript.

Response: Following the reviewer's suggestion, we have discussed the limited time points and underestimated of temporal variation in this study in lines 195-203, 259-269. As the reviewer mentioned environment variables explained twice of bacterial alpha diversity than fungi. This result suggests that fungal communities might be less variables to seasonal changes than bacteria. Also, we have now clarified that larger spatial distance for fungi vs. bacteria might be associated with the fact that bacteria might be affected by local changes in soil properties (Fierer and Jackson, 2006), while fungi is more affected by history processes like dispersal limitation and priority effect (Sun et al., 2017). This had been clarified in lines 236-252.

Reference:

- Fierer N, Jackson RB. 2006. The diversity and biogeography of soil bacterial communities. Proc Natl Acad Sci USA 103(3): 626-631.
- Sun S, Li S, Avera BN, Strahm BD, Badgley BD. 2017. Soil bacterial and fungal communities show distinct recovery patterns during forest ecosystem restoration. Appl Environ Microb 83: e00966-17.

The analysis of which microbial taxa differentiate communities (Fig.4) is only given one sentence in the results, and it is not included in the discussion at all. The significance of

this figure should be described further – possibly, are any of these taxa ecologically significant? Do these patterns confirm or refute other trends in the literature?

Response: Rapid changing bacteria Alphaproteobacteria and Gammaproteobacteria were the most sensitive to seasonal changes between winter and summer. Many of these taxa, often classified as copiotrophic organisms, had been demonstrated to be sensitive to the rapid change of soil moisture in previous study. The extended description and discussion have been added in lines 170-176, 207-218.

Line 119 – Were all community structure analyses performed with Bray-Curtis and Jaccard, and were the conclusions (qualitatively) the same? If so, consider making that more explicit.

Response: Yes, all the community structure analyses performed with Bray-Curtis and Jaccard and the conclusions were the same. More details have been added in lines 122-125.

Line 191 – This sentence includes undefined terms (such as legacy effect) and is repetitive.

“In fact, it’s reasonable the contemporary environmental factors only explained part of microbial spatial variation as the habitat type, space isolation and the legacy effect were likely regulate microbial spatial variations (14, 33). The large proportion of variations unexplained by season, spatial distance and environmental variables detected in our study might induced by the unmeasured environmental factors vary in space and time (14-15), biotic interaction (34-35) and ecological processes such as dormancy and persistence traits (6).”

This could be expanded on to discuss what types of biotic interactions may be relevant.

Response: The repetitive part has been deleted and the biotic interactions refer to the completion, mutualism and predation between microbial taxa and has been added in line 226.

Figure S5(b) – This figure text states that R2 for site is much higher than R2 for season, and that “Even though, the microbial community separated by winter and summer in each site, the differentiations were masked by sampling sites.” I think that merits a bit more explanation. Did this PCoA include environmental variables that could explain the grouping?

Response: The Figure S5 didn't include environmental variables. The points were grouped by season and sampling sites. Compared with Fig 1d, fungal community based on Jaccard distance really showed more clear seasonal variation. However, when considering the variation explained by site and season, the explanation ability of site was super larger than season based on both Jaccard and Bray-Curtis distance. The reason why we declared that microbial seasonal differentiations were masked by sampling sites was because microbial community showed clear separation in each site (Fig S7-S8). To make it clear, we added more details in lines 126-127.

Figures 1 and 3 should be more explicit about what the response variables and the predictors are.

Response: Revised in lines 528-532, 538-544.

Figure 3 is a rather confusing way to communicate that variance partitioning. A stacked bar plot may be more interpretable. If you proceed with a 4-way venn diagram, I recommend including more informative labels, and some emphasis on what parts of the diagram the reader is supposed to pay attention to.

Response: We have now changed the venn diagram to a stacked bar plot. Please see updated Fig.3.

Figure 4 lists many duplicate taxa, suggesting a problem with this figure. I know some names, like Actinobacteria, can refer to a phylum as well as a class, but

“Sordariomycetes” is repeated 9 times, and Gammaproteobacteria is repeated 6 times, and the legend claims that all are at the class level.

Response: Sorry for the lack of detail. Figure 4 shown the differential bacterial ESV and fungal ESV between winter and summer. The taxonomy information of each ESV assigned in Class level. That’s why the same names repeated many times. To make it clear, we have added the ESV ID in figure 4. Please see updated Fig. 4.

April 17, 2020

Prof. Haiyan Chu
Institute of Soil Science, Chinese Academy of Sciences
71 East Beijing Road
Nanjing 210008
China

Re: mSystems00783-19R1 (Spatial change-induced variability is more important than seasonality for shaping soil microbial spatiotemporal variation across a large spatial scale)

Dear Prof. Haiyan Chu:

Below you will find the comments of the reviewers. As you will see, the reviewers are positive about your revised manuscript, which can be accepted after you incorporate the requested minor textual revisions.

To submit your modified manuscript, log onto the eJP submission site at <https://msystems.msubmit.net/cgi-bin/main.plex>. If you cannot remember your password, click the "Can't remember your password?" link and follow the instructions on the screen. Go to Author Tasks and click the appropriate manuscript title to begin the resubmission process. The information that you entered when you first submitted the paper will be displayed. Please update the information as necessary. Provide (1) point-by-point responses to the issues raised by the reviewers as file type "Response to Reviewers," not in your cover letter, and (2) a PDF file that indicates the changes from the original submission (by highlighting or underlining the changes) as file type "Marked Up Manuscript - For Review Only."

Due to the SARS-CoV-2 pandemic, our typical 60 day deadline for revisions will not be applied. I hope that you will be able to submit a revised manuscript soon, but want to reassure you that the journal will be flexible in terms of timing, particularly if experimental revisions are needed. When you are ready to resubmit, please know that our staff and Editors are working remotely and handling submissions without delay.

To avoid unnecessary delay in publication, it is important that all elements you upload meet the technical requirements for production. I strongly recommend that you check your digital images using the Rapid Inspector tool at <http://rapidinspector.cadmus.com/RapidInspector/zmw/>.

Once your manuscript is accepted for publication, you will be contacted separately about payment when the proofs are issued; please follow the instructions in that e-mail. Arrangements for payment must be made before your article is published. For a complete list of **Publication Fees**, including supplemental material costs, please visit our website.

Sincerely,

Marnix Medema

Editor, mSystems

Journals Department
Reviewer comments:

Reviewer #1 (Comments for the Author):

Many thanks to the authors for addressing the comments one by one and considering the suggestions. The manuscript is improved very much and the quality of the work is more clear now. Now, only minor corrections in the text are required (see attachment).

Reviewer #2 (Comments for the Author):

This revision of the manuscript is substantially improved from the previous submission, and I believe it is a valuable and novel scientific contribution. There are still grammar/spelling mistakes, so I recommend that the editor work with the authors to improve readability of the text without changing the meaning.

Specific comments:

Although alpha and beta diversity are defined in the abstract, they are not defined in the Introduction or Results, which can make the results difficult to interpret.

Lines 119-125 - Confusing sentence organization, perhaps mention Bray-Curtis earlier in the sentence.

Line 129 - Unclear what the authors mean by "Combining microbial community similarity with geographic distance," but the change in turnover rates is very interesting and worth highlighting!

Line 132 - this is the first time "environmental distance" is used, should be defined.

Line 142 - "pH ... was the main factor involved in shaping both bacterial and fungal beta diversity" Could you report the direction of this relationship? Or does diversity for fungi/bacteria peak at specific pH values?

Paragraph from 144-155: When you say that environmental variables explain X amount of spatiotemporal variation, is it distance between environmental conditions that explains community difference? i.e. was every soil sample compared to every other sample, with the environmental/spatial/temporal distance as the predictors?

Line 166-169 - Confusing sentence, consider reorganizing to make it clearer which "factors figure

prominently in determining microbial seasonal variation"

Line 172 - The authors use "taxa," I assume instead of "ESVs", but I recommend clarifying that terminology or switching to "ESVs."

Line 198 - "Relic" DNA needs to be defined. Reference 11 does not appear to be the relic DNA study that is described in the text.

Figure 3 - I appreciate that the authors re-designed this variance partitioning figure, as the Euler diagram was confusing. However, the stacked bar plot completely obscures the overlapping nature of the variances. I strongly recommend exploring another option such as UpSet plots (via the UpSetR package <https://rdr.io/cran/UpSetR/>).

I have only some minor comments:

- The authors may re-consider to simplify the title
- 37-38: Please Re-check the grammar
- 41: "found" instead of "finding"
- 42-43: Re-writing
- 181: Typo
- 206: "implicate" instead of "implicates"
- Figure 1 and Figure S5: "number of ESVs" instead of "ESV"
- Typo in Figure 2: "similarity" "similarity" and "environmental distance" instead of "environment distance" (same for Figure S3)
- Figure 3: "Explained variation"

Dear Dr. Medema,

Thank you very much for the time you have spent on processing our manuscript (mSystems00783-19R1). We greatly appreciate the chance to address all comments from the editor and reviewers. Please find the attached updated manuscript and point-by-point response to the reviewers' comments.

Thank you for your time and attention!

Your sincerely,
Haiyan Chu
For the authors
April 21, 2020

Reviewer #1 (Comments for the Author):

Many thanks to the authors for addressing the comments one by one and considering the suggestion. The manuscript is improved very much and the quality of the work is more clear now. Now, only minor corrections in the text are required (see attachment).

Response: Thank you for your positive and constructive comments. We have carefully addressed the reviewer's concerns about typo and grammar mistakes. Please see our response to the Review Attachment.

Reviewer#1 Review attachment

I have only some minor comments:

The author may re-consider to simplify the title

Response: We have changed the title as "Space is more important than season when shaping soil microbial communities at large spatial scale". Please see lines 3-4.

37-38: Please Re-check the grammar

Response: We have modified the sentence in lines 37-38.

41: "found" instead of "finding"

Response: Revised in line 40.

42-43: Re-writing

Response: We have revised the sentence in lines 41-43.

181: Typo

Response: Revised in line 186.

206: "implicate" instead of "implicates"

Response: Revised in line 212.

Figure1 and Figure S5: "number of ESVs" instead of "ESV"

Response: Revised in Figure 1 and Figure S5.

Typo in Figure 2: “similarity” “similarity” and “environmental distance” instead of “environment distance” (same for Figure S3)

Response: Revised in Figure 2 and Figure S3.

Figure 3: “Explained variation”

Response: Revised in Figure 3.

Reviewer #2 (Comments for the Author):

This revision of the manuscript is substantially improved from the previous submission, and I believe it is a valuable and novel scientific contribution. There are still grammar/spelling mistakes, so I recommend that the editor work with the authors to improve readability of the text without changing the meaning.

Response: Thanks for the reviewer’s recognition of our work, we have carefully revised the grammar and spelling mistakes. Please see our response to the specific comments.

Specific comments:

Although alpha and beta diversity are defined in the abstract, they are not defined in the introduction or Results, which can make the results difficult to interpret.

Response: They had been defined in lines 117 and 121.

Line 119-125 - Confusing sentence organization, perhaps mention Bray-Curtis earlier in the sentence.

Response: Revised in lines 121-127.

Line 129 - Unclear what the authors mean by “Combining microbial community similarity with geographic distance,” but the change in turnover rates is very interesting and worth highlighting!

Response: We have modified the sentence for clarification, please see lines 131-134.

Line 132 - this is the first time “environmental distance” is used, should be defined.

Response: It has been well defined in lines 135-136.

Line 142 - “pH ...was the main factor involved in shaping both bacterial and fungal beta diversity” Could you report the direction of this relationship? Or does diversity for fungi/bacteria peak at specific pH values?

Response: Microbial beta diversity represents microbial composition similarity between sites, we can say that the sites with 2-unit pH difference have larger beta diversity than the sites with only 1-unit pH difference. So we didn't report if pH had positive or negative relationship with microbial beta diversity.

Paragraph from 144-155: When you say that environmental variables explain X amount of spatiotemporal variation, is it distance between environmental conditions that explains community difference? i.e. was every soil sample compared to every other sample, with the environmental/spatial/temporal distance as the predictors?

Response: Yes, the environmental/spatial/temporal distance were used as the predictors.

Line 166 - 169 - Confusing sentence, consider reorganizing to make it clearer which “factors figure prominently in determining microbial seasonal variation”

Response: We re-organized the sentence for clarification, please see lines 171-172.

Line 172 - The authors use “taxa”, I assume instead of “ESVs”, but I recommend clarifying that terminology or switching to “ESVs.”

Response: We switched it to ESVs in lines 31, 174, 176-178, 370, 372, 552, 554-555, 557.

Line 198 - “Relic” DNA needs to be defined. Reference 11 does not appear to be the relic DNA study that is described in the text.

Response: It has been defined in lines 203-204. Sorry for wrongly cited the number of the listed references. We have corrected it in line 206.

Figure 3 - I appreciate that the authors re-designed this variance partitioning figure, as the Euler diagram was confusing. However, the stacked bar plot completely obscures

the overlapping nature of the variances. I strongly recommend exploring another option such as UpSet plots (via the UpSetR package <https://rdrr.io/cran/UpSetR/>).

Response: As the reviewer suggested, we have modified the figure to UpSet plots. Please see Figure 3.

April 22, 2020

Prof. Haiyan Chu
Institute of Soil Science, Chinese Academy of Sciences
71 East Beijing Road
Nanjing 210008
China

Re: mSystems00783-19R2 (Space is more important than season when shaping soil microbial communities at large spatial scale)

Dear Prof. Haiyan Chu:

Your manuscript has been accepted, and I am forwarding it to the ASM Journals Department for publication. For your reference, ASM Journals' address is given below. Before it can be scheduled for publication, your manuscript will be checked by the mSystems senior production editor, Ellie Ghatineh, to make sure that all elements meet the technical requirements for publication. She will contact you if anything needs to be revised before copyediting and production can begin. Otherwise, you will be notified when your proofs are ready to be viewed.

Sincerely,

Marnix Medema
Editor, mSystems

Journals Department
Fig.S3: Accept

Table S1: Accept

Fig.S5: Accept

Fig.S8: Accept

Fig.S6: Accept

Fig.S7: Accept

Fig.S2: Accept

Table S2: Accept

Fig.S1: Accept

Fig.S4: Accept